# Continuous state-space representation of a bucket-type rainfall-runoff model: a case study with the GR4 model using State-Space GR4 (version 1.0)

Léonard Santos, Guillaume Thirel, and Charles Perrin

Irstea, UR HYCAR, 1 rue Pierre-Gilles de Gennes, 92160 Antony, France

*Correspondence to:* Léonard Santos (leonard.santos@irstea.fr)

**Abstract.** In many conceptual rainfall-runoff models, the water balance differential equations are not explicitly formulated. These differential equations are solved sequentially by splitting the equations into terms that can be solved analytically with a technique called "operator splitting". As a result, only the solutions of the split equations are used to present the different models. This article provides a methodology to make the governing water balance equations of a bucket-type rainfall-runoff model explicit and to solve them continuously. This is done by setting up a comprehensive state-space representation of the model. By representing it in this way, the operator splitting, which complexifies the structural analysis of the model, could be removed. In this state-space representation, the lag functions (unit hydrographs), which are frequent in rainfall-runoff models and make the resolution of the representation difficult, are first replaced by a so-called "Nash cascade" and then solved with a robust numerical integration technique. To illustrate this methodology, the GR4J model is taken as an example. The substitution of the unit hydrographs with a Nash cascade, even if it modifies the model behaviour when solved using operator splitting, does not modify it when the state-space representation is solved using an implicit integration technique. Indeed, the flow time series simulated by the new representation of the model are very similar to those simulated by the classic model. The use of a robust numerical technique that approximates continuous-time model also improves the lag parameter consistency across time steps and provides a more time-consistent model with time-independent parameters.

## 1 Introduction

### 1.1 On the need for an adequate mathematical and computational hydrological model

Hydrological modelling is a widely used tool to manage rivers at the catchment scale. It is used to predict floods and droughts as well as groundwater recharge and water quality. In a review on the different existing hydrological models, Gupta et al. (2012) determined that all the existing models follow three modelling steps:

- Establish a conceptual representation of reality,

- Represent this conceptualization in a mathematical model,

- Set up a computational model to be used on computer.

In terms of conceptual representation, many models exist and conceptualize the hydrological processes in the catchment differently, resulting in models with various levels of complexity. In this study, we will focus on the bucket-type models, which are among the simplest. These models, such as VIC (Wood et al., 1992), HBV (Bergström and Forsman, 1973) and Sacramento (Burnash, 1995), describe various conceptualizations of the hydrological processes at the catchment scale. Their parsimony

(they usually need few input data and use few parameters) make them very useful for research as well as in operational applications thanks to their robustness and good performance (Michel et al., 2006).

In the context of this study, bucket-type models are advantageous because, even if the concepts are often well documented, this is not the case of the mathematical and the computational models. In the models documentations, the water balance equations that would govern the models are rarely explicitly formulated (Clark and Kavetski, 2010). The authors of the models

often specify the discrete time equations, i.e. the result of the analytical or numerical temporal integration of the governing water balance equations. The problem is that the temporal resolution of the differential governing equations is part of the computational model. As a consequence, when the discrete time equations are the only ones available, the real mathematical model does not appear clearly. In addition, the descriptions of the numerical method used to solve the water balance equations and to obtain these discrete equations are rarely detailed.

However, several studies in the last decade (see for example Clark and Kavetski, 2010; Kavetski and Clark, 2010; Schoups et al., 2010) point out that the numerical solutions implemented to solve the differential equations that govern the models are sometimes poorly adapted. Clark and Kavetski (2010) showed that the use of the explicit Euler scheme (which is frequent for this type of model) can introduce significant errors in the simulated variables compared to more stable numerical schemes. Moreover, other studies prove that poorly adapted numerical treatment causes discontinuities and local optima in the parameter

hyperspace (Kavetski et al., 2003; Kavetski and Kuczera, 2007; Schoups et al., 2010). This results in problems efficiently calibrating the models and in uncertainty on parameter values.

Another numerical approximation is commonly applied for bucket-type models: the operator splitting (OS) technique (Kavetski et al., 2003). The aim is to split a differential equation into more simple equations that can be solved analytically in order to reduce inaccuracies in the numerical treatment. In the case of hydrological modelling, operator splitting results from

25 the sequential calculation of processes such as runoff, evaporation and percolation (Schoups et al., 2010). Kavetski et al. (2003), Clark and Kavetski (2010) and Schoups et al. (2010) identified several widely used models in which the differential equations are solved using this type of treatment, e.g. VIC (Wood et al., 1992), Sacramento (Burnash, 1995) and GR4J (Perrin et al., 2003). However, even if OS may reduce numerical errors, Fenicia et al. (2011) cite several limitations to its use in hydrology. Indeed, it is physically unsatisfying to separate the different processes in time because, in reality, they are concomittent. In

addition, it creates numerical splitting errors that are difficult to identify.

According to different studies, an inadequate numerical treatment like OS can lead to inconstencies in flux simulations (see for example the study conducted by Michel et al., 2003, on an exponential store). It may also create inconsistencies in the model state variables (Clark and Kavetski, 2010; Kavetski and Clark, 2010). This results in the model inaccurately simulating flows.

For these reasons, it is important to use a robust numerical treatment to better estimate the other uncertainties (for example, parameter uncertainty).

## 1.2 Scope of this study

The first step to improve the numerical treatment of rainfall-runoff models is to properly separate the mathematical model from the computational model (Kavetski and Clark, 2010; Gupta et al., 2012). This article proposes a method to do this by setting up a continuous state-space representation of a rainfall-runoff model. A state-space representation is a matricial function of a system that depends on input, output and state variables. At all times, the system is described by the values of its state variables (referred to as "states" in this article). In the case of rainfall-runoff models, inputs can be potential evapotranspiration and precipitation and output can be the flow at the outlet of the catchment. The soil water content or the amount of water in the hydrographic network are physical examples of possible state variables. The level of the bucket-type model stores is a conceptual example of possible state variables. This state-space representation will give the governing equations to be solved over time. This resolution will be proceeded by using an operator splitting technique to be used as a comparison point and by using a more robust numerical technique, *i.e.* implicit Euler with an adaptive sub-step number. The model solved by implicit Euler will be called continuous state-space because it approximates a continuous model. By opposition, the operator splitted state-space representation will be named as discrete.

In addition to a clearer mathematical model, we hope that the state-space representation will gain stability due to the direct implementation of the time step in the numerical resolution. We thus hope to obtain more stable parameter values across time steps (Young and Garnier, 2006).

To illustrate the methodology proposed, the widely used GR4J model (Perrin et al., 2003) will be taken as an example. Indeed, this model is currently implemented using the operator splitting technique. A state-space representation will be set up, following the GR4J's conceptualization of the hydrological processes as well as possible. Its behaviour, both with a discrete and a continuous solving, will be compared to the current formulation of the GR4J model on a wide range of French catchments with different time steps (day and hour), in terms of performance and parameters.

## 2 GR4 and its new state-space representation

Hereafter, the notation GR4 will be used to refer to structure of the GR4J model (J stand for *Journalier*, i.e. daily, Perrin et al., 2003), which is transformed and used at different time steps. This is a lumped bucket-type model discribed in its current form (Sect. 2.1) and in its state-space form (Sect. 2.2). A discussion on the Nash cascade introduced in the GR4 state-space form is given in Sect. 2.3. The continuous differential equations of the state-space form are described in Sect. 2.4. The adaptations needed to change the model time step will be described in Sect. 2.5.

## 2.1 Reference GR4 model

GR4 (Perrin et al., 2003) is a lumped bucket-type daily rainfall-runoff model with four free parameters. It is widely used for various hydrological applications in France (Grouillet et al., 2016; van Esse et al., 2013) and in other countries (Dakhlaoui et al., 2017; Seiller et al., 2017). It has shown good performances on a wide range of catchments (Coron et al., 2012). The equations of the reference GR4J model (Perrin et al., 2003) are the result of the integration of the water balance equations at a discrete time step (here the daily or hourly time step).

The version of GR4 used here is slightly different from the one presented by Perrin et al. (2003) because the two unit hydrographs were replaced by a single one placed before the flow separation (Fig. 1 (a), Mathevet, 2005). This simplification of the model does not substantially change the resulting simulated flows.

The equations of the model are given by Perrin et al. (2003) and listed in Table 1. GR4 represents the rainfall-runoff relationship at the catchment scale using an interception function, two stores, a unit hydrograph and an exchange function (see Fig. 1 (a)). The model structure can be split into water balance and routing operators.

The water balance operators evaluate effective rainfall (i.e. the part of rainfall that will reach the catchment outlet) by estimating several quantities: actual evaporation, storage within the catchment and groundwater exchange. It involves an interception function and a production (soil moisture accounting) store ($S$ in Fig. 1 (a)). The interception corresponds to a neutralization of rainfall by potential evapotranspiration. The remaining rainfall ($P_n$), if any, is split into a part going into the production store ($P_s$ in Fig. 1 (a)) and a complementary part ($P_n - P_s$ in Fig. 1 (a)) that is directed to the routing component of the model. The quantity of rainfall that feeds the production store depends on the level of water in the store at the beginning of the time step. In case there is remaining energy for evapotranspiration after interception ($E_n$ in Fig. 1 (a)), some water is evaporated from the production store at an actual rate depending on the level of the production store ($E_s$ in Fig. 1 (a)). The higher the level is at the beginning of the time step, the closer $E_s$ is to $E_n$. Thus, the production store represents the evolution of the catchment moisture content at each time step. The last water balance operator is a groundwater exchange term ($F$ in Fig. 1 (a), positive or negative), which acts on the routing part of the model.

The routing function of the model is fed with the rainfall that does not feed the production store ($P_s - P_n$) plus a percolation term ($Perc$ in Fig. 1 (a)) from the production store, which generally represents a small amount of water. The total amount ($P_r$ in Fig. 1 (a)) is lagged by a symmetric unit hydrograph and then split into two flow components. The main component (90% of $P_r$, $Q_9$ in Fig. 1 (a)) is routed by a nonlinear routing store ($R$ in Fig. 1 (a)). The complementary component (10% of $P_r$, $Q_1$ in Fig. 1 (a)) directly reaches the outlet. The groundwater exchange term ($F$) is added or removed from the routing store and from the $Q_1$ component.

The simulated flow at the catchment outlet ($Q$ in Fig. 1 (a)) is the sum of the outputs of the two flow components ($Q_r$ and $Q_d$ in Fig. 1 (a)).

Four free parameters (called $x_1$, $x_2$, $x_3$ and $x_4$) are used to adapt the model to the variety of catchments. Their meanings are given in Table 2.

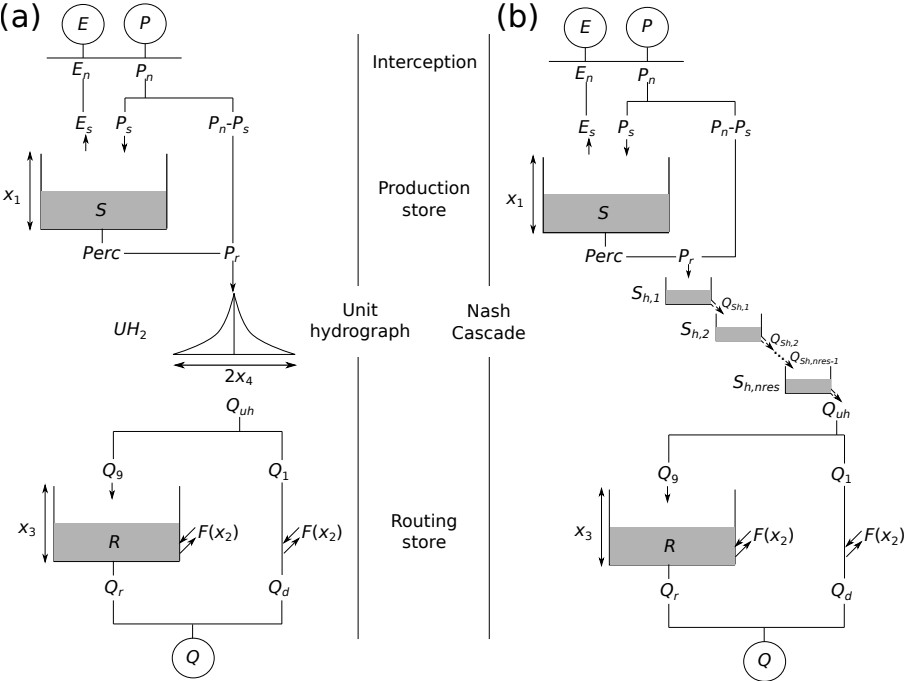

**Figure 1.** Schemes of the reference GR4 model ((a), Perrin et al., 2003) and the state-space (b) structures. $P$: rainfall; $E$: potential evapotranspiration; $Q$: streamflow; $x_i$: model parameter; other letters are model state variables or fluxes. A Nash cascade replaces the unit hydrograph in the state-space representation.

As mentioned in the introduction, the governing water balance equations of the model are solved using operator splitting. By considering that inputs to the store are added at the beginning of the time step as Dirac functions (Michel, 1991), it becomes possible to find analytical expressions of the model processes when equations are integrated over the time step. Consequently, the model processes are treated sequentially.

## 2.2    A state-space formulation for the GR4 model

5  To create this state-space representation, it is important to identify the different model state variables. In the GR4 model, two obvious states are the levels of the production and routing stores. The main challenge to describe the state-space formulation is to deal with the unit hydrograph. The discrete form used in GR4 corresponds to a convolution product in the state space as implemented in SUPERFLEX (Kavetski and Fenicia, 2011). This convolution product complexifies the mathematical resolution of the model that is necessary for the continuous version that will be introduced in Sect. 2.4. Here we chose to replace this

10  unit hydrograph with a series of linear stores in order to simplify this resolution. The use of stores is also convenient because it creates a model that is only composed of stores.

Different combinations of linear stores were tested and the choice was made to replace the unit hydrograph with a "Nash cascade" (Nash, 1957). It is implemented at the same location in the model structure as the unit hydrograph (Fig. 1 (b)). The "Nash cascade" is a chain of linear stores that empty into each other. It has two parameters to govern the shape of the outflow response, namely the number of stores and the outflow coefficient, which is identical for all stores. In our case, we decided to

fix the number of stores and to only consider the outflow coefficient as a free parameter. This choice will be discussed in the following section (Sect. 2.3). With this type of model, the outflow of the last store has a similar shape to a unit hydrograph.

## 2.3 Parameterisation of the Nash cascade

As introduced in the previous section, the Nash cascade has two parameters, namely the number of stores and the outflow coefficient. The number of stores can only take integer values, which is an issue for automatic calibration because it introduces

threshold effects. As a consequence, the number of stores was not optimized automatically and the outflow coefficient is the preferential parameter to calibrate.

To obtain a response that is equivalent to the GR4 unit hydrograph response, we attempted to determine whether a relationship exists between the Nash cascade parameters and the GR4 $x_4$ parameter. To manage this, the determination of the Nash cascade parameter is based on the comparison of the impulse response of the Nash cascade and the response of the unit

hydrograph.

The impulse response of the Nash cascade is (Nash, 1957):

$$h_{Nash}(t) = \frac{k}{\Gamma(nres)} (kt)^{nres-1} \exp(-kt) \tag{1}$$

where $h_{Nash}(t)$ is the impulse response of the Nash cascade at time $t$, $nres$ is the number of stores, $k$ is the outflow coefficient (in $t^{-1}$) and $\Gamma(nres)$ corresponds to the gamma function of $nres$.

The impulse response of the GR4 symmetrical unit hydrograph is (Perrin et al., 2003):

$$h_{UH}(t) = \begin{cases} \frac{2.5}{2x_4} \left(\frac{t}{x_4}\right)^{1.5} & , \text{for } 0 \leqslant t \leqslant x_4 \\ \frac{2.5}{2x_4} \left(2 - \frac{t}{x_4}\right)^{1.5} & , \text{for } x_4 < t \leqslant 2x_4 \\ 0 & , \text{for } t > 2x_4 \end{cases} \tag{2}$$

where $h_{UH}(t)$ is the impulse response of the unit hydrograph at time $t$, $x_4$ is the time to peak of the hydrograph.

The Nash cascade parameters are calculated depending on $x_4$ in such a way that the time to peak and the peak flow would be the same for the two impulse responses. According to Szöllösi-Nagy (1982), the time to peak of the Nash cascade is equal

to:

$$t_p = \frac{nres - 1}{k} \tag{3}$$

and the peak flow is equal to:

$$q_p = \frac{k}{\Gamma(nres)} (nres - 1)^{nres-1} \exp(1 - nres) \tag{4}$$

Using Eq. 2, the time to peak of the GR4 unit hydrograph is equal to:

$$t_p = x_4 \tag{5}$$

and the peak flow to:

$$q_p = \frac{1.25}{x_4} \tag{6}$$

So, from these values the following system can be deduced:

$$
\begin{cases}
x_4 & = & \frac{nres-1}{k} \\
\frac{1.25}{x_4} & = & \frac{k}{\Gamma(nres)}(nres-1)^{nres-1}\exp(1-nres)
\end{cases} \tag{7}
$$

which can be transformed into:

$$
\begin{cases}
k & = & \frac{nres-1}{x_4} \\
1.25 & = & \frac{(nres-1)^{nres}}{\Gamma(nres)}\exp(1-nres)
\end{cases} \tag{8}
$$

A number of stores $nres = 11$ is the best integer approximation to solve the second equation of Eq. 8. The outflow coefficient is deduced from this number of stores and from $x_4$. By fixing the parameters in this way, only the $x_4$ parameter has to be calibrated. This method allows a direct comparison between the parameters of the Nash cascade and the parameter of the unit hydrograph. For a given $x_4$ parameter, the unit hydrograph and the Nash cascade impulse responses have the same time to peak and the same peak flow (see the dotted and the dashed curve in Fig. 2).

Using this formula, the $x_4$ parameters of the two models are equivalent and it can be argued that their meaning is nearly identical.

Fixing the number of stores in the Nash cascade also provides another advantage. Indeed, one of the potential issues that arise when replacing the unit hydrograph with a Nash cascade was the equifinality with the routing store. Given that the recession curve of the cascade is theoretically infinite, it could have the same function as the routing store. Calculating the parameters of the cascade regarding the $x_4$ parameter makes it possible to reduce the possibility of an infinite impulse response.

## 2.4 Continuous differential equations of the state-space model

Once the model is only represented by stores, a differential equation can be written for each store (details are provided in Table 1). For the production and routing stores, the equations were built by adding all the processes that affect the stores. For example, the differential equation for the production store is the sum of the differential equations of evaporation, rainfall and the percolation (respectively, $E_s$, $P_s$ and $Perc$ in Fig. 1). This means that all the processes that are a function of this state are treated simultaneously, unlike the initial model version in which the processes are treated sequentially. The state-space representation of the Nash cascade is the same as the one proposed by Szöllösi-Nagy (1982).

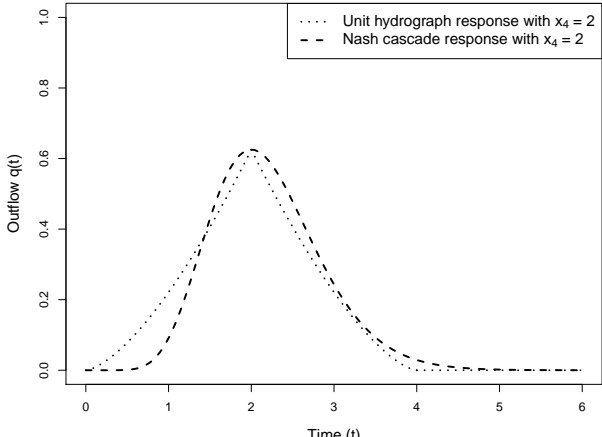

**Figure 2.** Impulse response with a $x_4 = 2$ time steps for the unit hydrograph of GR4 (dotted line) and the Nash cascade with $nres = 11$ stores and $k = \frac{11-1}{x_4}$ (dashed line).

The resulting model is composed of the differential equations governing the states' evolution (here represented as a vector in the Eq. 9, taking into account $nres$ stores in the Nash cascade):

$$
\begin{pmatrix}
\dot{S} \\
\dot{S}_{h,1} \\
\dot{S}_{h,2} \\
\vdots \\
\dot{S}_{h,nres} \\
\dot{R}
\end{pmatrix}
=
\begin{pmatrix}
P_s - E_s - Perc \\
P_r - Q_{Sh,1} \\
Q_{Sh,1} - Q_{Sh,2} \\
\vdots \\
Q_{Sh,nres-1} - Q_{uh} \\
Q_9 + F - Q_r
\end{pmatrix}
\tag{9}
$$

The notation $\dot{S}$ stands for $\frac{dS}{dt}$, the derivative of $S$ against time $t$ and the different elements of this equation are specified in Table 1.

The output equation to calculate the instantaneous output flow ($q(t)$ in Eq. 10) completes the model:

$$
q(t) = Q_r + Q_d
\tag{10}
$$

The different elements in Eq. 9 and 10 are shown in Table 1.

The input, state variable and output values are:

– Inputs: $E_n$ and $P_n$ are the potential evapotranspiration (after the interception) and the precipitation amounts after the

interception phase in $\mathrm{mm \cdot t^{-1}}$. We decided to keep the interception out of the state-space representation, because it is not represented by a store in the reference GR4J and we wanted to avoid introducing an additional difference between the state-space and the reference models.

- Output: $Q$ is the output flow, it corresponds to the integration of $q(t)$ (Eq.10) over the time step.

- State variables: $S$, $R$ and $S_{h,k}$ are respectively the levels of the production store, the routing store and the Nash cascade store number $k$ (with $k \in \{1, \cdots, nres\}$) in mm.

- Fluxes: $P_s$ and $E_s$ are, respectively, the rainfall added to the production store and the evapotranspiration extracted from the production store. $Perc$ is the outflow from the production store. $P_r$ is the amount of water that reaches the model routing operators. $Q_{Sh,k}$ is the outflow of the Nash cascade store number $k$ (with $k \in \{1, \cdots, nres-1\}$). $Q_{uh}$ is the outflow of the Nash cascade store number $nres$ (this notation is used to be consistent with the discrete model). $Q_9$ and $Q_r$ are, respectively, the inflow and the outflow of the routing store and $F$ is the inter-catchment groundwater exchange. $Q_d$ is the outflow of the complementary flow component.

The parameter meanings are explained in Table 2. The model is constructed to ensure that the parameters ($x_1, \cdots, x_4$ in the equations) have the same meaning in the continuous model and in the discrete GR4. The state-space formulation was sought to be as close as possible to the original model's formulation, to keep the same general model structure. We expect similar results to be obtained by the different tested model versions.

## 2.5 Hourly model

The GR4 model was first designed for daily time step modelling and it was adapted for the hourly time step (GR4H, Mathevet, 2005; Ficchí et al., 2016). The structure and the equations are similar in GR4H (hourly) and in GR4J (daily). The hourly versions of the GR4 models used here are the same as the ones showed in Fig. 1.

The adaptation to the time step is handled by a change in the parameter values, which depend on time. Ficchí et al. (2016) gave the theoretical relationships to transform the GR4 free parameter values as a function of the time step length (Table 3). The fixed percolation coefficient ($\nu$ in Table 1) is also time-dependent.

The continuous state-space GR4 model used for the hourly time step is exactly the same as the one used at the daily time step, with no change in the percolation coefficient. The time step change is not managed by a change in parameter values but by the numerical integration. For the daily time step, the model is integrated on $\Delta t = 1 \, \mathrm{day}$ while, for the hourly time step, it is integrated on $\Delta t = 1 \, \mathrm{hour}$.

## 3 Implementation and testing methodology

### 3.1 Numerical integration of the model

The integration of Eq. 9 (necessary to adapt the model to discrete input data) cannot be made analytically. It is therefore necessary to implement a numerical method to solve this integration.

**Table 1.** Details of the equations of the GR4 model, discrete and continuous formulations. The discrete formulations are the continuous equations integrated individually over the modelling time step using the operator splitting technique while continuous equations correspond to the terms of the water balance differential equation of each store. (*) The values of $UH_2$ are calculated using Eq. (17) in Perrin et al. (2003). Please note that the two discrete formulations use either the Unit hydrograph equations or the Nash cascade formulation.

| Model component name | Notation | Flux name | Discrete formulations | Continuous formulation |
|---|---|---|---|---|
| Production store | S | Precipitation in the store | $P_s = \dfrac{x_1\left(1-\left(\frac{S}{x_1}\right)^\alpha\right)\tanh\frac{P_n}{x_1}}{1+\frac{S}{x_1}\tanh\frac{P_n}{x_1}}$ | $P_s = P_n\left(1-\left(\frac{S}{x_1}\right)^\alpha\right)$ |
| | | Evaporation from the store | $E_s = \dfrac{\left(2S-\frac{S^\alpha}{x_1}\right)\tanh\frac{E_n}{x_1}}{1+\left(1-\frac{S}{x_1}\right)\tanh\frac{E_n}{x_1}}$ | $E_s = E_n\left(2\frac{S}{x_1}-\left(\frac{S}{x_1}\right)^\alpha\right)$ |
| | | Percolation | $Perc = S\left(1-\left(1+\left(\nu\frac{S}{x_1}\right)^{\beta-1}\right)^{\frac{1}{1-\beta}}\right)$ | $Perc = \dfrac{x_1^{1-\beta}}{(\beta-1)U_t}\nu^{\beta-1}S^\beta$ |
| Unit Hydrograph | $UH_2$ | UH inflow | $P_r = P_n - P_s + Perc$ | - |
| | | UH outflow | $Q_{uh} = P_r * UH_2^{(*)}$ (convolution product) | |
| Nash cascade | $S_{h,1}$ | Precipitation inflow in store 1 | $P_r = P_n - P_s + Perc$ | $P_r = P_n - P_s + Perc$ |
| | | Store 1 outflow | $Q_{Sh,1} = S_{h,1}\left(1-\exp\left(\frac{1-nres}{x_4}\right)\right)$ | $Q_{Sh,1} = \frac{nres-1}{x_4}S_{h,1}$ |
| | $S_{h,2}$ | Store 2 inflow | $Q_{Sh,1}$ | $Q_{Sh,1}$ |
| | | Store 2 outflow | $Q_{Sh,2} = S_{h,2}\left(1-\exp\left(\frac{1-nres}{x_4}\right)\right)$ | $Q_{Sh,2} = \frac{nres-1}{x_4}S_{h,2}$ |
| | $\cdots$ | $\cdots$ | $\cdots$ | $\cdots$ |
| | $S_{h,n}$ | Store nres inflow | $Q_{Sh,nres-1}$ | $Q_{sh,nres-1} = \frac{nres-1}{x_4}S_{h,nres-1}$ |
| | | Store nres outflow | $Q_{uh} = S_{h,nres}\left(1-\exp\left(\frac{1-nres}{x_4}\right)\right)$ | $Q_{uh} = \frac{nres-1}{x_4}S_{h,nres}$ |
| Routing store | R | Routing store inflow | $Q_9 = \Phi Q_{uh}$ | $Q_9 = \Phi Q_{uh}$ |
| | | Inter-catchment exchanges | $F = \frac{x_2}{x_3^\omega}R^\omega$ | $F = \frac{x_2}{x_3^\omega}R^\omega$ |
| | | Routing store outflow | $Q_r = R\left(1-\left(1+\left(\frac{R}{x_3}\right)^{\gamma-1}\right)^{\frac{1}{1-\gamma}}\right)$ | $Q_r = \frac{x_3^{1-\gamma}}{(\gamma-1)U_t}R^\gamma$ |
| Output flow | $Q = Q_r + Q_d$ | Routing store outflow | $Q_r$ | $Q_r$ |
| | | Direct flow | $Q_d = max(0;(1-\Phi)Q_{uh}-F)$ | $Q_d = max(0;(1-\Phi)Q_{uh}-F)$ |

**Table 2.** Meaning of the free and fixed parameters (from Perrin et al., 2003, except for $U_t$ and $nres$)

| Type | Name | Signification | Value | Unit |
|------|------|---------------|-------|------|
| Free | $x_1$ | Max capacity of the production store | - | mm |
| | $x_2$ | Inter-catchment exchange coefficient | - | $\mathrm{mm \cdot t^{-1}}$ |
| | $x_3$ | Max capacity of the routing store | - | mm |
| | $x_4$ | Base time of the unit hydrograph | - | t |
| Fixed | $\alpha$ | Production precipitation exponent | 2 | - |
| | $\beta$ | Percolation exponent | 5 | - |
| | $\gamma$ | Routing outflow exponent | 5 | - |
| | $\omega$ | Exchange exponent | $\frac{7}{2}$ | - |
| | $\epsilon$ | Unit hydrograph coefficient | $\frac{3}{2}$ | - |
| | $\Phi$ | Partition between routing store and direct flow | 0.9 | - |
| | $\nu$ | Percolation coefficient | $\frac{4}{9}$ | - |
| | $U_t$ | One time step length | 1 | t |
| | $nres$ | Number of stores in Nash cascade | 11 | - |

**Table 3.** Temporal transformations of the GR4 parameters (Ficchí et al., 2016)

| GR4 model parameter | Theoretical transformation from the daily ($\Delta t_d$) to the hourly ($\Delta t_h$) time step | Source of time step dependency |
|---------------------|------------------------------------------------------------------------------|--------------------------------|
| $\nu$ | $\nu_{\Delta t_h} = \nu_{\Delta t_d} \left( \frac{\Delta t_d}{\Delta t_h} \right)^{\frac{1}{4}}$ | Integration of the percolation power 5 function from the production store |
| $x_1$ | $x_{1(\Delta t_h)} = x_{1(\Delta t_d)}$ | - |
| $x_2$ | $x_{2(\Delta t_h)} = x_{2(\Delta t_d)} \left( \frac{\Delta t_d}{\Delta t_h} \right)^{-\frac{1}{8}}$ | Integration of the exchange flux formulation (dependent on the routing store level) |
| $x_3$ | $x_{3(\Delta t_h)} = x_{3(\Delta t_d)} \left( \frac{\Delta t_d}{\Delta t_h} \right)^{\frac{1}{4}}$ | Integration of the fueling power 5 function of the routing store |
| $x_4$ | $x_{4(\Delta t_h)} = x_{4(\Delta t_d)} \left( \frac{\Delta t_d}{\Delta t_h} \right)$ | Discrete concentration time in time step units of the unit hydrographs |

Following the recommendation in Clark and Kavetski (2010), an implicit Euler algorithm is used to perform this numerical integration. Our choice was to set up an adaptive sub-step algorithm (Press et al., 1992) to avoid the majority of numerical errors. The implicit equation is solved using a secant method when necessary.

The choice of using adaptive sub-step rather than single-step implicit method (as recommended by Clark and Kavetski, 2010) is a result of several tests that are not shown here. We compared the modelling results with single-step integration to those obtained with the adaptive sub-step algorithms and found some differences in resulting flows (in particular for high flows). The differences found this way were not negligible. In this case, we can say that the stability of the implicit single-step integration is not sufficient to sufficiently reduce the integration errors.

For both hourly and daily time steps, the inputs are considered as constant during the time step. Even if this assumption is a simplification of the truth, we chose to keep it constant to simplify the calculation and not to introduce treatment differences between hourly and daily time step models.

## 3.2 Catchment set and data

To compare the performance and behaviour of the reference and the discrete and continuous state-space GR4 model versions, a large data set of 240 catchments across France was set up (Fig. 3). Testing the models on many catchments will help obtain general conclusions (Andréassian et al., 2006; Gupta et al., 2012).

The data set was built by Ficchí et al. (2016) to test GR4 at different time steps. In this article, we only used daily and hourly data. The climate data of the SAFRAN daily reanalysis (Quintana Seguì et al., 2008; Vidal et al., 2010) are used as input data (precipitation and temperature). Precipitation and temperature are spatially aggregated on each catchment since the GR4 models are lumped. The hourly precipitation data were obtained by disaggregating the daily SAFRAN precipitation using the subdaily distribution of rain gauge measurements. Potential evapotranspiration at the daily time step was calculated from the SAFRAN temperature using the Oudin formula (Oudin et al., 2005) and hourly spread with a Gaussian distribution. Full details on this data set are available in Ficchí et al. (2016).

Hourly observed flows are available at each catchment outlet and come from the *Banque HYDRO* (http://www.hydro.eaufrance.fr/, French Ministry of the Environment). For daily modelling, hourly measurements are aggregated at the daily time step. Their availability covers the 2003-2013 period.

The catchments were selected to have less than 10% precipitation falling as snow, to avoid requiring a snow model.

## 3.3 Testing methodology

Three versions of the model were assessed on the 240 catchments following a split-sample test (Klemeš, 1986). These three versions are the reference model, a discrete state-space model (with a Nash Cascade but solved using operator splitting) and a continuous state-space model. Comparing the reference and discrete state-space models allows to measure the impact of replacing the unit hydrograph with a Nash cascade. Comparing the discrete and continuous state-space models allows to measure the impact of a nearly continuous numerical integration. For every catchment, the observed flow data period was divided into a calibration period (the first half) and a validation period (the second half). A 2-year warm up period was used for

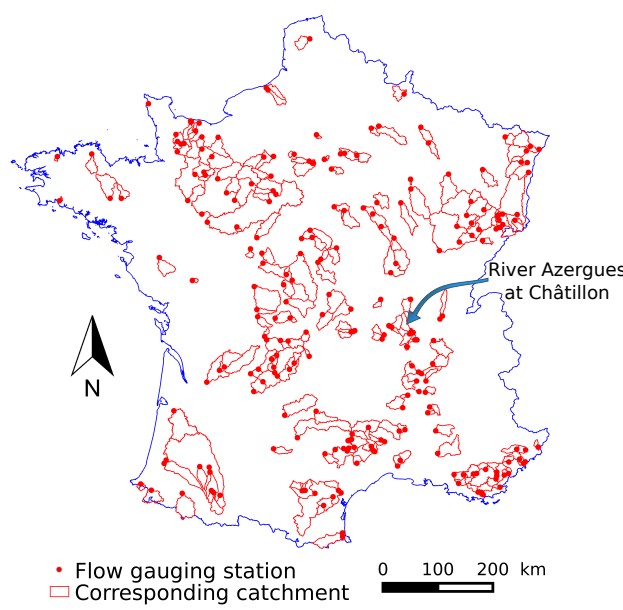

**Figure 3.** Location of the 240 flow gauging stations used for the tests and their associated catchments. The River Azergues at Châtillon is used as an example for the results (Sect. 4.1).

each catchment, before both the calibration and validation periods. The calibration was made automatically with an algorithm used in Coron et al. (2017) and based on the work of Michel (1991).

The objective function used for calibration is the Kling-Gupta Efficiency (KGE', Kling et al., 2012). This objective function is often used in hydrology and assesses different components of the error made by the model (mean bias, variance bias, correlation). In addition, to target different flow levels, mathematical transformations are applied (Pushpalatha et al., 2012). The logarithm is applied to analyse the errors in low-flow conditions ($KGE'\left(\log\left(Q\right)\right)$), no transformation is applied to preferentially analyse the error on high flows ($KGE'\left(Q\right)$) and the root square of the flow is used as a compromise representing the error on intermediate flows ($KGE'\left(\sqrt{Q}\right)$). In the case of logarithm transformation, following the recommendations made by Pushpalatha et al. (2012), a small quantity which corresponds to one hundredth of the catchment mean flow is added to avoid troubles with null flows. These three transformations represent three distinct objective functions. The models were calibrated separately and successively on the three objective functions. To avoid strongly negative values of the KGE' criterion, we used the $C_{2M}$ formulation which restricts the variation range into $[-1; 1]$ (see Mathevet et al., 2006).

The results of the calibrations were also analysed in terms of performance in validation on the three evaluation criteria (i.e. $C_{2M}\left(Q\right)$, $C_{2M}\left(\log\left(Q\right)\right)$ and $C_{2M}\left(\sqrt{Q}\right)$). Given the large number of catchments, it is possible to draw a conclusion on the global difference in performance between the three studied model versions. This avoids a discrepancy due to specific catchment conditions. In addition to the performance analysis, the simulated hydrographs were visually analysed to detect discrepancies in the flow simulation. An analysis of the time series of internal fluxes and state variables also provided further

insights to interpret the difference between the model versions. Last, the differences in parameter values between the models was analysed. It is important to verify that the parameter values are similar and do not take outlier values that would compensate for model inconsistencies.

A second test was carried out in order to analyse the time step dependency of the models. The split-sample test was performed
at the hourly time step and the parameter values were compared to those obtained at the daily time step. With the reference model, the calibrated parameter values were compared to those theoretically obtained using the equations in Table 3. With the continuous state-space model, we verified the stability of the parameters. This stability is very important for designing a model that is not dependent on its time step.

## 4   Results and discussion

### 4.1   Comparison of tested models at the daily time step

Figure 4 shows that performances are globally similar between the different versions of the model with a calibration using the $C_{2M}$ on square-rooted flows. The performances of the reference model and the continuous state-space solution are also similar after calibration with the two other transformations of the flow in the objective function (not shown). In the case of the discrete state-space solution, the model does not seem to be able to well reproduce high flows but performs better on low flows than the
two other models when the used objective function is the $C_{2M}$ with logarithmic transformation.

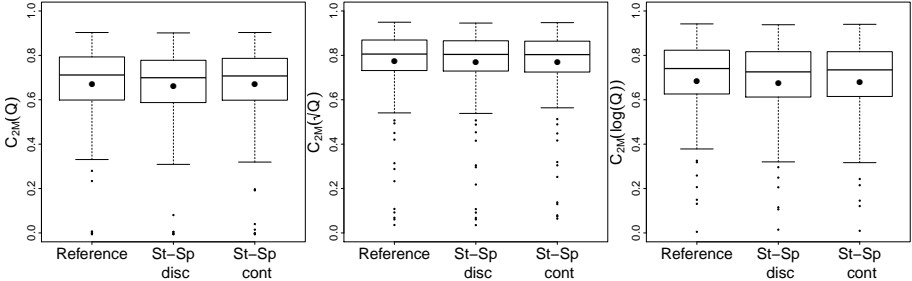

**Figure 4.** Performance comparisons obtained in validation between the reference (with unit hydrograph), the discrete state-space (with Nash cascade) and the continuous state-space daily GR4, on 240 catchments, focusing on high (left), intermediate (middle) and low (right) flows after calibration with the $C_{2M}\left(\sqrt{Q}\right)$ (i.e. focusing on intermediate flow). The large points represent the mean performance and the smaller ones represent the outliers. The 5, 25, 50, 75 and 95 percentiles are represented by the boxplots.

The study of the hydrographs provides complementary information. The reference GR4 model and the continuous state-space solution are very similar while the discrete state-space solution simulates lower peak flows (see example hydrograph in Fig. 5). This behaviour can be explained because solving the eleven linear stores introduces errors that propagate and amplify across the Nash cascade.

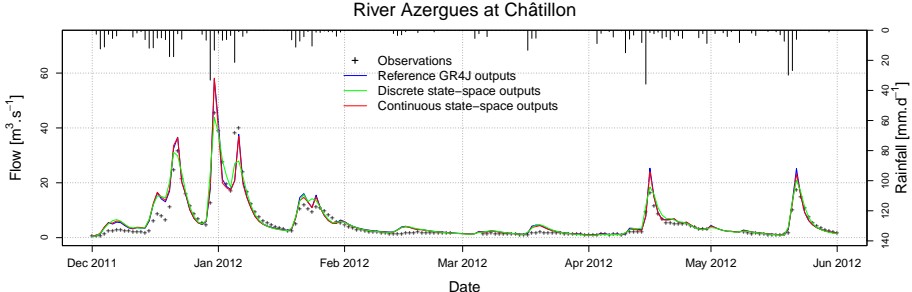

**Figure 5.** Simulated hydrograph of the River Azergues in the first half of 2012 during the validation period. The reference GR4 model (output in blue), the GR4 discrete state-space solution (output in green) and the continuous state-space solution (output in red) were calibrated with $C_{2M}\left(\sqrt{Q}\right)$ as the objective function.

To extend the analysis on the similarity of the models, we compared the parameter values obtained by calibration. As shown in Fig. 6, the parameters have the same range of values. We still can note differences in the values of the $x_4$ parameter, which are systematically higher for the discrete state-space model. These differences in the values are probably due to the differences in response shape between the Nash cascade and the unit hydrograph (see Sect. 2.3) and to the errors produced by operator splitting solving of the Nash cascade. The assumption that the differences in $x_4$ values are due to errors caused by unsuitable solving is confirmed by the fact that the $x_4$ parameter values are similar for the three models at hourly time step (not shown here).

Last, to understand the internal impact of the state-space formulation on the model, we analysed state variables and internal fluxes. Two differences are induced by the model's state-space formulation. First, the discrete Nash cascade output peaks are lower than the peaks of the unit hydrograph (Fig. 7). The peaks of the continuous state-space representation are more similar with the reference but the peaks occur sooner. The second difference between the models concerns the levels of the routing store (Fig. 8). Here we only compared the reference GR4 to the continuous state-space solution because the input in the routing store are too different for the discrete state-space solution. The peak levels are higher in the continuous state-space representation, even sometimes higher than the maximum capacity of the routing store. The reason for this is that we shifted from the discrete model in which the processes are treated sequentially to a continuous model in which all the processes are solved simultaneously. In the discrete model, the exchanges are first calculated based on the routing level at the beginning of the time step, then the output of the unit hydrograph is added and last the outflow of the routing store is calculated. Due to this sequential treatment, in high-flow conditions, the quantity of exchanged water and the outflow of the routing store in the discrete model is lower than those of the continuous state-space representation. Given that most of the time the exchange parameter is negative, the lower outflow of the routing store is compensated by less water loss with the groundwater exchange in the complementary flow branch. This can explain why the simulated flows are similar despite these internal differences.

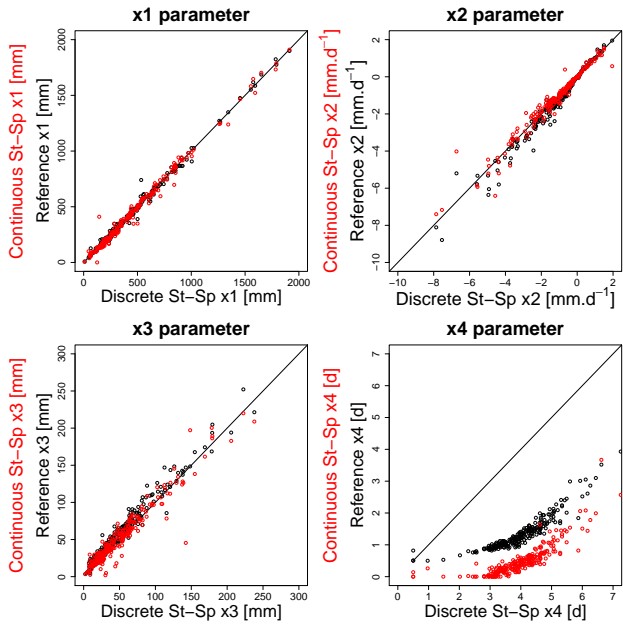

**Figure 6.** Scatter plots of the four free parameters of the different versions of the models obtained by calibration with $C_{2M}\left(\sqrt{Q}\right)$ as an objective function on the basins of the data set. Parameter comparison between unit hydrograph and Nash cascade is in black and parameter comparison between discrete and continuous state-space parameters is in red. The values of $x_1$, $x_2$ and $x_3$ are similar for the models (the line represents the $y = x$ line). The $x_4$ values are higher in the discrete state-space model than for the other model versions.

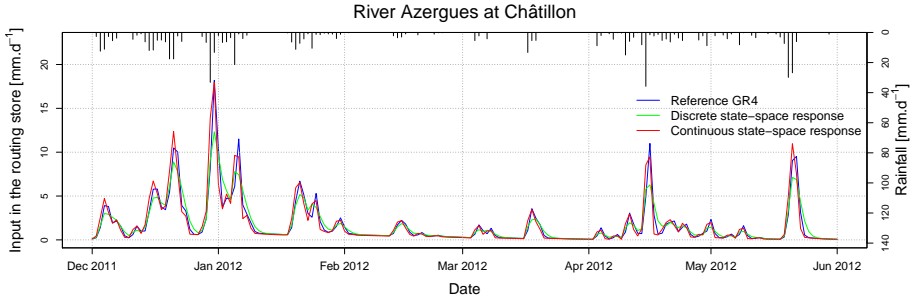

**Figure 7.** Daily inputs in the routing store of the River Azergues in the first half of 2012. The models are calibrated with the $C_{2M}\left(\sqrt{Q}\right)$ as the objective function. The peaks are lower with the discrete state-space GR4 (green lines) and occur sooner with the continuous state-space GR4 (red lines).

Moreover, by analysing the differences between the two models, it is also important to take into account the computational time. Indeed, running the original model version is on average three times faster than the continuous state-space version due to
5    the adaptive sub-step method. This is important to consider for some applications.

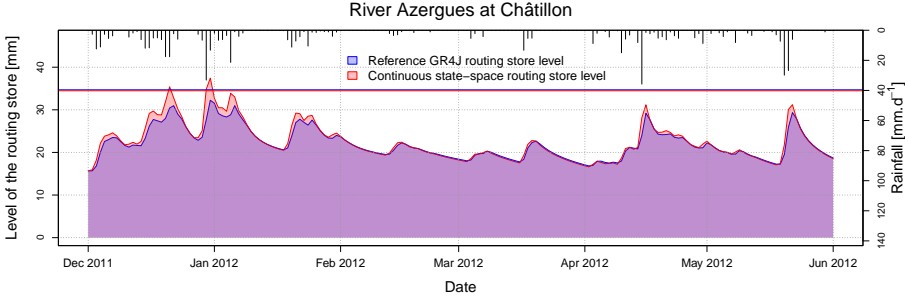

**Figure 8.** Daily routing store filling of the River Azergues in the first half of 2012. The reference GR4 (blue line) and the continuous state-space representation (red line) are calibrated with the $C_{2M}\left(\sqrt{Q}\right)$ as the objective function.

This computational time rise is essentially due to the adaptive sub-step algorithm. For example, in the River Azergues at Châtillon catchment, the mean number of sub-steps is 22 and it can reach 100 during some days. However, in Sect. 3.1 we argue that the adaptive sub-step method seems necessary to avoid numerical errors.

To conclude with these results, we can argue that the modifications brought by the continuous state-space representation, although they modify the model's internal fluxes, do not degrade the model's performance, but only slightly modify the model's internal fluxes. It is important to underline that the operator splitting solving of a Nash cascade creates more errors than a discrete unit hydrograph. To be equivalent to the reference model, the state-space representation of GR4 needs to be solved with a robust numerical technique.

## 4.2 Consistency of the state-space representation through time steps

The analysis of temporal consistency provides the most valuable result produced by the continuous state-space representation. The work of Ficchí et al. (2016) resulted in a GR4 model that is nearly consistent across time steps. However, to adapt the model, they chose to include the time step variations in a theoretical transformation between the free parameter values and the percolation fixed coefficient (Table 3) at different time steps. In this section, we only compare the reference GR4 with the continuous solution of the state-space representation. The parameters of the state-space representation discrete solution show the same behaviour as the reference GR4 ones so it was chosen not to show them. This proves that all the improvements shown in this section are only due to the continuous resolution of the state-space model.

In Fig. 9, the free parameter values obtained by calibration at the hourly time step are compared to those obtained at the daily time step using the reference GR4 version. The dashed lines represent the regression obtained by the theoretical relations reported in Table 3. One can note that the calibrated parameters (the dots in Fig. 9) are quite different between the two time steps but it is important to note that the values of the $x_3$ parameter follow the relations proposed by Ficchí et al. (2016) (the dashed lines). The high values of $x_1$ are underestimated compared to the theoretical relation as are the low values of the $x_2$ parameter. There is also an issue with the unit hydrograph parameter ($x_4$ in Fig. 9) for which calibrated hourly parameter values

are systematically lower than the values it would have by following the transformation. Kavetski et al. (2011) and Littlewood and Croke (2008) encountered the same issue with the lag parameter of their models.

30     The values of $x_1$, $x_2$ and $x_4$ are inconsistent compared to the values expected using the theoretical transformations. Regarding the work of Ficchí (2017), we can argue that the changes in the high values of $x_1$ and the low values of $x_2$ are due to temporal inconsistencies in the interception calculation. The case of the $x_4$ parameter is more problematic. The differences in the $x_4$ values probably stem from the discretization of the unit hydrograph at different time steps.

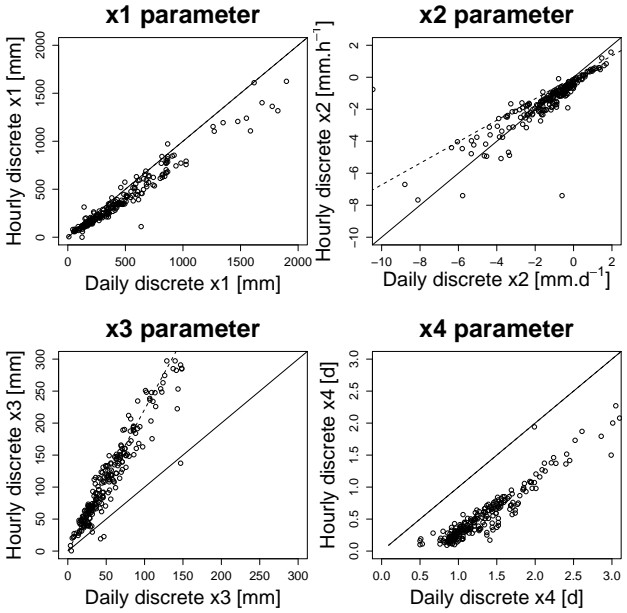

**Figure 9.** Scatter plots representing the four parameters of the reference (daily and hourly) GR4 models obtained by calibration with $C_{2M}\left(\sqrt{Q}\right)$ as objective function. The solid line represents the $y = x$ regression and the dashed lines the transformation relations of Table 3.

    In the continuous state-space model, the time step is taken into account in the temporal numerical integration of the model. For this reason, in theory there is no need to adapt the values of the parameters. This is confirmed in Fig. 10, where the values of calibrated parameters remain approximately constant despite the time step change. Only the high values of $x_1$ and the values of $x_2$ slightly diverge from the $x = y$ line.

    This result is useful in building a model that can adapt its time step resolution depending on given conditions. The results

5  are particularly interesting for the case of $x_4$ values because the $x_4$ values are constant between the two time steps, resolving the issue encountered by Littlewood and Croke (2008), Kavetski et al. (2011) and Ficchí et al. (2016) with lag parameters. As explained in the work of Littlewood and Croke (2013), this improvement can be explained by the fact that the adaptive sub-step integration approximates a continuous time input in the Nash cascade. The results obtained with the $x_4$ parameter

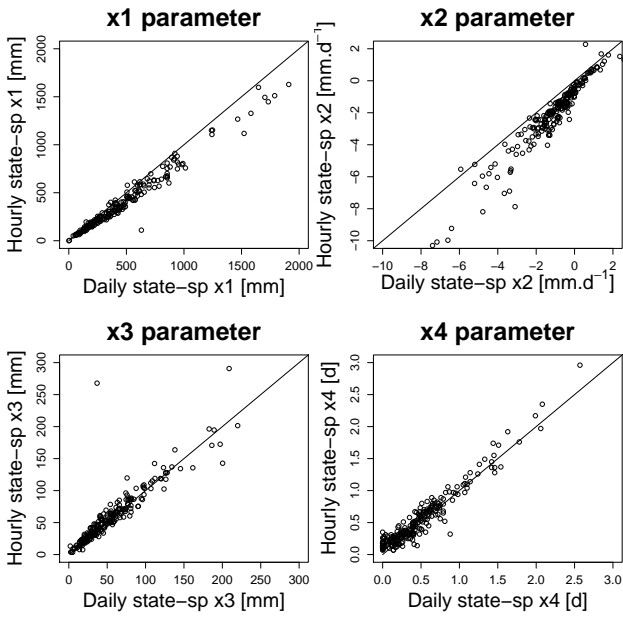

**Figure 10.** Scatter plots representing the four parameters of the continuous state-space (daily and hourly) GR4 models obtained by calibration with $C_{2M}\left(\sqrt{Q}\right)$ as the objective function. The solid line represents the $y = x$ line.

here tend to confirm on a wide range of catchments this earlier work. However, in addition to the input errors, the lack of $x_4$
10  time consistency can also be explained by the integration errors produced by the operator splitting at daily time step.

The outliers in $x_3$ values that occur in Fig. 10 are also present in Fig. 9. No explanations relating to physical characteristics of these catchments or simulation performance were found. We assume that these outliers values are due to the non sensitivity of the $x_3$ parameter for these catchments.

Finally, to verify stability, we also need to compare the performance of the models at the hourly time step. Figure 11 shows that, as at the daily time step, the performance is similar for the different versions.

Thus, the continuous state-space representation shows better temporal stability in the $x_4$ parameter values with similar performance.

## 5   Conclusions and perspectives

5  The objective of this study was to present a version of a bucket-type rainfall-runoff model with a robust numerical resolution of the governing water balance equations by setting up a continuous state-space representation. The methodology is based on (i) identifying the state variables, (ii) writing their differential equations, (iii) replacing certain components of the model with more easily described components in terms of differential equations (namely replacing the unit hydrograph with a Nash cascade here), (iv) solve these equations with a robust numerical integration technique. Finally, all the fluxes that form the water

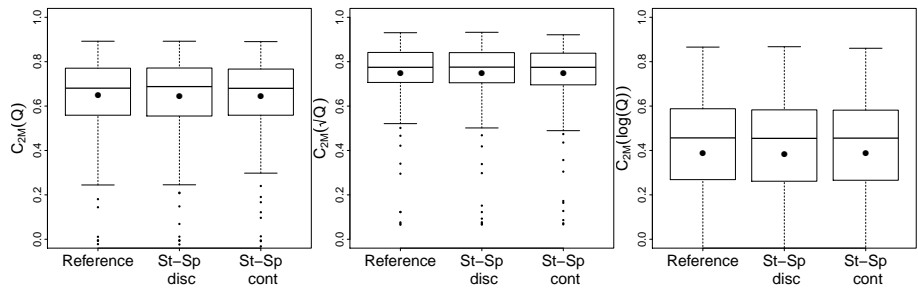

**Figure 11.** Performance comparisons obtained in validation between the reference (with unit hydrograph), the discrete state-space (with Nash cascade) and the continuous state-space hourly GR4, on 240 catchments, focusing on high (left), intermediate (middle) and low (right) flows after calibration with the $C_{2M}\left(\sqrt{Q}\right)$ (i.e. focusing on intermediate flow). The points represent the mean performance.

5    balance equation governing a state are solved simultaneously while they are solved sequentially in operator splitted models. As stated by Fenicia et al. (2011), this is more physically satisfying.

This work was presented using the example of the GR4 model. The new version was created to be as close as possible to the initial model but a single modification was implemented: a Nash cascade substitutes the model's unit hydrograph.

When analysing the results and the output flows, it was shown that the new formulation, when solved with a robust numerical 10    technique, has a limited impact on performance. However, the analysis of the parameter values and of the internal fluxes of the model shows that some discrepancies occur when running the model. The peak flow of the Nash cascade occurs sooner than the peak flow of the unit hydrograph. The amount of water in the routing store and exchanged by the groundwater exchange function is also higher for the state-space representation, particularly during high-flow periods.

Nonetheless, the continuous state-space representation simulates flows that are very similar to the flows simulated by the 15    original GR4 version and performs equally well. It also seems to provide greater stability in the parameter values, particularly regarding different modelling time steps. Moreover, the use of the Nash cascade rather than the unit hydrograph improves (when solved with implicit Euler) the lag parameter value stability with time steps. This improved stability can make it easier to calibrate the model with a given data set and to apply it at a finer time step for which no discharge data are available. It can also allow using a model that runs at a finer time step in high-flow periods and a larger time step in low-flow periods.

20    Furthermore, the comparison between the discrete and continuous state-space model shows that the benefits provided by the continuous state-space representation are a result of the use of a robust numerical integration technique. Indeed, solving the state-space representation using operator splitting introduces errors that impact the simulated flow values and do not result in parameter stability. Thus, the real benefit of the use of the Nash cascade is to simplify the numerical solving application.

The performance obtained with the continuous state-space model is not better than that of the original model. In addition, 25    because the number of sub-steps sometimes needs to be high, the computational time is longer with the continuous state-space representation of the model. Consequently, the use of this representation would be helpful for particular applications such as time-variable modelling. It might also be useful for certain data assimilation techniques (typically variational methods) because all the components are represented as states and the governing equations are clearly defined.

In addition, it could also be advantageous to find a way to adapt the number of stores of the Nash cascade to the catchment studied.

Although it is necessary to adapt the Nash Cascade to different unit hydrograph shapes, this article suggests a sufficiently general methodology to erase operator splitting in hydrological bucket-type modelling and can be transposed to other models.

## 6 Code and data availability

The Fortran code used in this article can be freely downloaded from GitHub at:

https://github.com/HYDRO-group-Irstea-Antony/GR4-State-space-version-1.0

The state-space model can be tested on an example catchment data set with already calibrated model parameters. The full reference for this code can be found in the references (Santos, 2017), it is referenced with the following doi:

https://doi.org/10.5281/zenodo.1118183.

*Author contributions.* This work is part of L. Santos' PhD work, he made the technical development, the analysis and wrote the manuscript. G. Thirel and C. Perrin are the PhD supervisors, they supervised this work and the manuscript writing.

*Competing interests.* The authors declare that they have no conflicts of interest.

*Acknowledgements.* The first author's PhD grant was provided by Irstea. We thank Météo France for providing the SAFRAN climatic data used in this work. We also would like to thank Martyn Clark for his advice in setting-up the differential equations, Nicolas Le Moine for sharing his ideas to replace the unit hydrographs and on the numerical integration, Fabrizio Fenicia for his advice on numerical integration and Paul-Henry Cournède for his analysis of the mathematical adequacy of the model. Finally, we give special thanks to Andrea Ficchí for his work on the database and for the discussions on the temporal stability of the GR4 model.

We thank the topical editor, Dr Jeffrey Neal, for his monitoring of the review process and his relevant reviewers choice. We also acknowledge the two reviewers, Dr Barry Croke and an anonymous reviewer for their very interesting and complementary remarks.

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
