# Peer review of "Continuous state-space representation of a bucket-type rainfall-runoff model: a case study with the GR4 model using State-Space GR4 (version 1.0)"

_Geoscientific Model Development, 2017_

## Short Comment (SC1) · 13 Dec 2017

GMD is encouraging authors to provide a persistent access to the exact version of the source code used for the model version presented in the paper. As explained in https://www.geoscientific-model-development.net/about/manuscript_types.html the preferred reference to this release is through the use of a DOI which then can be cited in the paper. For projects in GitHub a DOI for a released code version can easily created using Zenodo, see https://guides.github.com/activities/citable-code/ for details. Please note that in the code accessibility section you can still point the reader to the GitHub repository for the newest version even if you use a DOI for the relevant release.

[Figure]

Lutz Gross GMD Executive Editor

---

## Author Comment (AC1) · 19 Dec 2017

We would like to thank the GMD executive editor Prof. Lutz Gross for his comment. As suggested, we archived the source code on Zenodo and created the following doi: https://doi.org/10.5281/zenodo.1118183.

Please note that this doi point to a github account created for the research group the authors of the manuscript belong to, not to the personal account of Léonard Santos, who is a PhD student. That way, the access to further evolutions of the code developed by our group is warranted.

[Figure]

The link to the github repository will be amended in a further version of the manuscript and the following citation to the software will be cited in the code availability section and added in the reference list:

Léonard Santos. (2017, December 19). HYDRO-group-Irstea-Antony/GR4-State-space-version-1.0: First release of GR4-State-space-version-1.0 (Version v1.0). Zenodo. http://doi.org/10.5281/zenodo.1118183

Léonard Santos, on behalf of co-authors

---

## Referee Comment (RC1) · B. Croke (Referee) · 5 Jan 2018

**General Comments**

This paper describes a continuous time state space formulation of a revised GR4J model, using a Nash Cascade for the Unit Hydrograph module. The state space formulation is then compared with the discrete time version of the model. This procedure itself is not new, but application to a complete hydrological model is. The result is interesting: no significant change in the performance of the model, but significantly less variation in the parameter values for hourly and daily time steps with the state space model (figure 9) compared to the discrete time model (figure 8). For applications

where the interpretation of the parameter values is important (e.g. regionalisation), the state space model may give better result. The paper is very well written. The paper would be of interest to model developers across disciplines, and would be a worthwhile contribution to the literature.

**Specific comments**

1. Page 9, section 3.1: The issue is not really instability but rather the size of the error in the approximation given by the numerical method. There is instability when the errors grow with time. Yes, this is definitely a problem, but the problem starts before this point. Even if the errors decay with time (resulting in a stable solution), they can be large enough to cause problems, particularly if the decay is sufficiently slow. What is needed is a numerical method that gives a sufficiently small error at the timestep of interest. The reason for going to a finer sub-step is to reduce the error in the numerical approximation, not to avoid instability (essentially, stability is a necessary but not sufficient condition). This is a flaw that exists in the literature, but it would be good to not continue to propagate it. Another point here is that by going to a sub-step calculation, you are making assumptions about how the inputs (rainfall and potential evaporation) are distributed within a time step. Is the rainfall a delta function at the start of the time step, a constant rate over the time step (zero order hold), or something else?

2. Page 10, section 3.3, line 9: Given the use of a log transform, are there zero flows present, or are all stream perennial? If there are zero flows, how are these handled? Options are to simply ignore them (meaning the model can take any value for time steps with zero flow), or use the two parameter Box-Cox transformation. The later should be generally preferred as this includes assessment of the performance of the model even when the observed flows are zero.

3. Page 14, line 22-24: This may be due to the sub-step calculation in the numerical integration. This would convert the model to something approaching a

continuous time model (using the zero order hold), as in the papers published by Littlewood, Croke and Young (2011; HSJ, 56:3, 521-524) and Littlewood and Croke (2013; Hydrology Research, 44, 430-440). These papers compared a discrete-time model (IHACRES) and a continuous-time model (CT-DBM model in the Captain toolbox), and showed that the variation in the parameter values was significantly smaller for the continuous-time model. This re-emphasizes the need for the distribution of the climate input within a sub-step to be defined.

4. Page 16, Figure 9: There are a couple of outliers in the x3 plot, one with an extremely large difference in the value. Any ideas why this catchment is behaving so differently? Is it a very small catchment?

5. Page 16, Figure 10: Obviously there are same extremely large negative values in the KGE values using log transformed flows. This means that some of the models are giving very poor fits. Presumably the mean value for the state space model is just a little below zero? Might be worth including a little more discussion on this?

6. Page 17, line 7: Not really correct to say that $nres$=11 solves the second equation in equation 10. $nres$=11 gives a value of 1.2511, so it approximates the required value of 1.25 very closely, but doesn't solve it.

**Typographic errors**

1. Caption for figure 1: "discrete"

2. Page 17, equation 10: error in superscript in second equation (has $(nres-1)^n res$ rather than $(nres-1)^{nres}$).

---

## Referee Comment (RC2) · Anonymous Referee #2 · 24 Jan 2018

The paper implements a state-space representation of the popular rainfall-runoff model GR4. The state-space representation allows solving the model using robust numerical techniques as opposed to adhoc operator splitting techniques used in the old version of the model. A novel finding in this paper is that the new model version results in more robust parameter estimates that are less sensitive to the temporal resolution (hourly vs daily) of the model simulations. The paper is well written and the results are interesting. I hope my comments below help the authors to further improve their paper:

1. In developing the state-space representation of their model, the authors introduce two changes. First, a different routing model is used (Nash cascade vs unit hydro-

[Figure]

graph). And second, the model is solved with a different numerical technique (implicit Euler with adaptive time stepping vs operator splitting approach with fixed time step). It would be preferable to introduce these two changes separately rather than together, so as to separate the effects of these two changes.

2. Run times are longer with the new model compared to the original implementation due to the use of implicit Euler with adaptive time stepping. Have you considered using a single-step implicit Euler integration? This may be faster without losing the benefits of the new implementation.

3. Please provide some details/examples of the actual time steps and number of non-linear iterations in your model, for example for one specific basin.

4. Questions about the state-space formulation, Eq. 1:

- why not include water balance of the interception store as an additional differential equation?

- Simulated discharge Q in Eq.2 is defined as an instantaneous flow I assume? Observed discharge is however an integrated quantity (total over an hour or a day). Wouldn't it be better to define simulated Q also as an integrated quantity? You could in fact add Eq.2 to the ODE system in Eq. 1: $dQ/dt = Q_r + Q_d$. Note that you then would have to reset $Q = 0$ at the start of each forcing time interval.

- it would be good to explicitly point out in table 1 that the instantaneous flux equations are the same for the two models.

5. Section 2: The discrete form is contrasted with the state-space form of the model. Note that a state-space representation can be either discrete or continuous, so it may be better to explicitly call it continuous state-space formulation.

6. Section 4.3: this section describes the relation between the unit hydrograph approach for routing in the old model and the Nash cascade representation in the new model; in my view this section really fits better in the methods section, for example

following the text at the bottom of page 6. My suggestion is to move it there.

7. Abstract: what do you mean by "resolution"?

8. Typos:

p4 line 20: symetric -> symmetric

caption fig.1: discret -> discrete

p9 line 2: adaptative -> adaptive

p11 line 6: tose -> those

p14 line 16: unconsistencies -> inconsistencies

p18 line 20: grounwater -> groundwater

caption fig.11: comparaison -> comparison

---

## Author Response (AR1)

**Answer to the review comments by Dr Barry Croke**

We would like to thank Dr Barry Croke for his detailed analysis and suggestions on the article. They will help improving the quality of the manuscript.

**Specific comments**

1. Page 9, section 3.1: *The issue is not really instability but rather the size of the error in the approximation given by the numerical method. There is instability when the errors grow with time. Yes, this is definitely a problem, but the problem starts before this point. Even if the errors decay with time (resulting in a stable solution), they can be large enough to cause problems, particularly if the decay is sufficiently slow. What is needed is a numerical method that gives a sufficiently small error at the time-step of interest. The reason for going to a finer sub-step is to reduce the error in the numerical approximation, not to avoid instability (essentially, stability is a necessary but not sufficient condition). This is a flaw that exists in the literature, but it would be good to not continue to propagate it. Another point here is that by going to a sub-step calculation, you are making assumptions about how the inputs (rainfall and potential evaporation) are distributed within a time step. Is the rainfall a delta function at the start of the time step, a constant rate over the time step (zero order hold), or something else?*

    We agree that stability is necessary but not sufficient. However, the adaptive sub-step calculation method used in this work is designed to particularly reduce instabilities as the sub-step value calculation is based on the difference between two consecutive solutions obtained with different tested sub-step values. To be sure that this method was interesting we compared it to a fixed sub-step Euler implicit method with one hundred sub-steps and the differences between the two were very low. Regarding the second remark on the need not to propagate the confusion between error and instabilities, we will better explain this point in the revised version of the article. About the assumption on the input distribution, we considered the input as constant over a time-step (over one day for the daily model and over one hour for the hourly model). We are aware that this is a simplification of the truth but without more indications at the sub-hourly time-step we decided to keep it constant for the hourly and daily models. We will add this information in the section 3.1 to help understanding.

    **Added/Modified:** Sect. 3.1, p 12, line 4 (of the revised manuscript), *The choice of using adaptive sub-step rather than single-step implicit method (as recommended by Clark and Kavetski, 2010) is a result of several tests that are not shown here. We compared the modelling results with single-step integration to those obtained with the adaptive sub-step algorithms and found some differences in resulting flows (in particular for high flows). The differences found this way were not negligible. In this case, we can say that the stability of the implicit single-step integration is not sufficient to sufficiently reduce the integration errors.*

    *For both hourly and daily time steps, the inputs are considered as constant during the time step. Even if this assumption is a simplification of the truth, we chose*

*to keep it constant to simplify the calculation and not to introduce treatment differences between hourly and daily time step models.*

2. Page 10, section 3.3, line 9: Given the use of a log transform, are there zero flows present, or are all stream perennial? If there are zero flows, how are these handled? Options are to simply ignore them (meaning the model can take any value for time steps with zero flow), or use the two parameter Box-Cox transformation. The later should be generally preferred as this includes assessment of the performance of the model even when the observed flows are zero.

To handle the zero flow, a small quantity corresponding to one hundredth of the mean flow of the catchment is added to flow in the log transform. This technique was used by Pushpalatha et al. (2012) on the Nash-Sutcliffe efficiency and we adopted it. This will be specified in the revised version of the manuscript.

**Added/Modified:** Sect. 3.3, p 13, line 8, *In the case of logarithm transformation, following the recommendations made by Pushpalatha et al. (2012), a small quantity which corresponds to one hundredth of the catchment mean flow is added to avoid troubles with null flows.*

3. Page 14, line 22-24: This may be due to the sub-step calculation in the numerical integration. This would convert the model to something approaching a continuous time model (using the zero order hold), as in the papers published by Littlewood, Croke and Young (2011; HSJ, 56:3, 521-524) and Littlewood and Croke (2013; Hydrology Research, 44, 430-440). These papers compared a discrete-time model (IHACRES) and a continuous-time model (CT-DBM model in the Captain toolbox), and showed that the variation in the parameter values was significantly smaller for the continuous-time model. This re-emphasizes the need for the distribution of the climate input within a sub-step to be defined.

This is a very good remark, the production store differential equation resolution can approximate a continuous time runoff input as used with CT-DBM in the 2013 Hydrology Reasearch article. Regarding this approximation, it tends to confirm on a wide range of catchments the result that this paper highlighted. However, we can also explain the difference between $x_4$ parameters by the higher errors due to operator-splitting approximation in differential equations resolution at daily time-step. The higher errors may introduce differences in calibrated parameter values. This is, in our opinion, a combination of these two modifications that allow the parameters values to be constant across time-steps. In this context, we can admit that the constant distribution of input is problematic but, until now, it is the best approximation that we can use. We will further discuss this point in the article and introduce the cited references.

**Added/Modified:** Sect. 4.2, p 18, line 16, *As explained in the work of Littlewood and Croke (2013), this improvement can be explained by the fact that the adaptive sub-step integration approximates a continuous time input in the Nash cascade. The results obtained with the $x_4$ parameter here tend to confirm on a wide range of catchments this earlier work. However, in addition to the input errors, the lack of $x_4$ time consistency can also be explained by the integration errors produced by the operator splitting at daily time step.*

4. Page 16, Figure 9: There are a couple of outliers in the x3 plot, one with an extremely large difference in the value. Any ideas why this catchment is behaving so differently? Is it a very small catchment?

   Indeed, the two outliers catchments are small catchments (145 and 20 square kilometers area). But, as other studied catchments with a similar area did not face this issue, it is not the only reason to explain this behaviour. This is neither due to the state-space transformation because the parameter differences between daily and hourly transformation also exist with the discrete version of the model for these catchments nor is it due to performances because models are quite good on these catchments. The difference between daily and hourly parameter values may be due to $x_3$ parameter insensitivity on these catchments. We will discuss the case of these outliers in the article.

   **Added/Modified:** Sect. 4.2, p 19, line 3, *The outliers in $x_3$ values that occur in Fig. 10 are also present in Fig. 9. No explanations relating to physical characteristics of these catchments or simulation performance were found. We assume that these outliers values are due to the non sensitivity of the $x_3$ parameter for these catchments.*

5. Page 16, Figure 10: Obviously there are same extremely large negative values in the KGE values using log transformed flows. This means that some of the models are giving very poor fits. Presumably the mean value for the state space model is just a little below zero? Might be worth including a little more discussion on this?

   The mean KGE' on the log is -0.0825, this negative value is due to some strongly negative KGE' values. To deal with these values that introduce troubles in performances analysis, we will replace the KGE' criteria used in the article by a bounded version of it. This version, bounded between $-1$ and $1$, is calculated like the $C_{2M}$ criterion (Mathevet et al., 2006; IAHS Publ. 307; 211-219) which is based on Nash-Sutcliffe efficiency. The formulation will be:

$$C_{2M} = \frac{KGE'}{2 - KGE'} \tag{1}$$

   **Added/Modified:** Sect. 3.3, p 13, line 11, *To avoid strongly negative values of the KGE' criterion, we used the $C_{2M}$ formulation which restricts the variation range into $[-1; 1]$ (see Mathevet et al., 2006).*

   *We also modified all the figures and occurrences of KGE' and replaced it by the $C_{2M}$.*

6. Page 17, line 7: Not really correct to say that $nres = 11$ solves the second equation in equation 10. $nres = 11$ gives a value of 1.2511, so it approximates the required value of 1.25 very closely, but doesn't solve it.

   Indeed, 11 is the integer that gives the best approximation for the equation 10. Thus, we chose this integer as the number of stores in the Nash Cascade. We will be more precise in the sentence by writing: "A number of store $nres = 11$ is the best integer approximation to solve the second equation of Eq. 10"

**Added/Modified:** Sect. 2.3, p 7, line 9, *A number of stores $nres = 11$ is the best integer approximation to solve the second equation of Eq. 8.*

Typographic errors will also be corrected.
**Added/Modified:** *Done*

Léonard Santos, on behalf of co-authors

**Answer to the review comments of Reviewer #2**

We would like to thank the reviewer for his analysis and suggestions on the article which are, in our opinion, complementary to those made by the other reviewer, Dr Barry Croke.

1. In developing the state-space representation of their model, the authors introduce two changes. First, a different routing model is used (Nash cascade vs unit hydrograph). And second, the model is solved with a different numerical technique (implicit Euler with adaptive time stepping vs operator splitting approach with fixed time step). It would be preferable to introduce these two changes separately rather than together, so as to separate the effects of these two changes.

   We thank the reviewer for this remark. We made additional tests to investigate this, we replaced the unit hydrograph by a Nash Cascade but integrated it using operator-splitting. This replacement does not change the performances and, when using hourly time step, the parameters values are similar for the two operator-splitted models. However, at the daily time-step, the $x_4$ parameter values of the Nash Cascade are higher than the ones of the unit hydrgraph at daily time-step. It tends to prove that the insensitivity of the $x_4$ parameter values to temporal resolution (highlighted in the section 4.2 of the article) is not due to the replacement of the unit hydrograph by a Nash Cascade. These remarks will be taken into consideration in the revised version of the article.

   **Added/Modified:** *This remark induced various modifications in the text, in the conclusion and in the abstract. By introducing the changes separately, we found that the insensitivity of parameter to temporal resolution is essentially due to the use of a robust numerical integration technique.*

2. Run times are longer with the new model compared to the original implementation due to the use of implicit Euler with adaptive time stepping. Have you considered using a single-step implicit Euler integration? This may be faster without losing the benefits of the new implementation.

   Even if it is not mentioned in the article, we tested the Implicit Euler method with increasing sub-steps number from 1 to 100. The number of sub-steps seems to have an influence, particularly in high flow periods. To illustrate the impact of using a single-step, we compare (see Fig. 1 below) the boxplots of performance for an adaptive sub-step number implicit integration and a single-step Euler implicit one. The GR4 parameters used for this comparison are the ones obtained by the GR4 calibration on KGE' calculated on square rooted streamflows that is presented in the article. The boxplots show a decrease of performances. This tends to show that, even if single-step implicit Euler does not face instabilities when solving the equations, it can increase errors. This can be linked to the second comment made by Dr Barry Croke.

   Another important disadvantage of not using sub-stepping is that it does not solve the parameter time instability issue. To prove it, we calibrated the continuous state-space model at the daily and hourly time-steps using a single-step implicit

[Figure]

Figure 1: Performances comparisons between adaptive sub-step and single-step Implicit Euler methods

Euler method without sub-steps. In the Fig. 2, we plotted the resulting parameter scatter plots comparison (the same way that is used in Fig. 9 in the article). Unlike with adaptive time-step, the $x_4$ parameters show differences between daily and hourly time-steps. This result tends to confirm Barry Croke's third remark in which he argues that increasing the sub-step number can help to approach a continuous time model.

[Figure]

Figure 2: Scatter plots representing the four parameters of the discrete (daily and hourly) GR4 with Nash Cascade models obtained by calibration with $KGE'(\sqrt{Q})$ as the objective function. The solid line represents the $y = x$ line.

**Added/Modified:** Sect. 3.1, p 12, line 3 (of the revised manuscript), *The choice of using adaptive sub-step rather than single-step implicit method (as recommended by Clark and Kavetski, 2010) is a result of several tests that are not shown here. We compared the modelling results with single-step integration to those obtained with the adaptive sub-step algorithms and found some differences in resulting flows (in particular for high flows). The differences found this way were not negligible. In this case, we can say that the stability of the implicit single-step integration is not sufficient to sufficiently reduce the integration errors.*

3. Please provide some details/examples of the actual time steps and number of non-linear iterations in your model, for example for one specific basin.

   If we take the example of the River Azergue at Chatillon catchment (the example catchment chosen in the article) on the validation period, the mean number of used sub-steps is 2 for hourly simulation and 22 for daily simulation. Figure 3 graph shows the cumulative appearance frequency of the different numbers of sub-steps. It is, in majority, one or two sub-steps for the hourly time-step but it is more variable in the case of daily simulation.

[Figure]

Figure 3: Cumulative appearance frequency of the number of sub-steps obtained with adaptive Implicit Euler resolution of the continue GR4 state-space model at the daily and hourly time-steps

   At the hourly time-step, we found out that the number of sub-steps increases when the rainfall amount increases. In the case of daily time-step it is not clear, possibly because the number of sub-steps is correlated with a combination of rainfall and the stores levels. We can notice that the average daily sub-step value (which approximately corresponds to 1 hour) is higher than the average hourly sub-step value (approximatively 0.5 hour). This is probably due to the fact that the maximum sub-step value for the hourly simulation is limited to 1 hour. We will make a comment on this observation in the article.

   **Added/Modified:** Sect. 4.1, p 17, line 6, *This computational time rise is essentially due to the adaptive sub-step algorithm. For example, in the River Azergues at Châtillon catchment, the mean number of sub-steps is* 22 *and it can reach* 100 *during some days. However, in Sect. 3.1 we argue that the adaptive sub-step method seems necessary to avoid numerical errors.*

4. Questions about the state-space formulation, Eq. 1:

   - Why not include water balance of the interception store as an additional differential equation?

In the current version of GR4, the interception is not calculated with a store but it is a simple difference between rainfall and potential evapotranspiration. Only one input (which is the difference between the larger and the smaller of the two) is considered in the model, which is a difference with other bucket-type rainfall-runoff models. We decided not to change this input calculation in order not to include more differences between the two models. This answer will be added to the article.

**Added/Modified:** Sect. 2.4, p 8, line 25, *We decided to keep the interception out of the state-space representation, because it is not represented by a store in the reference GR4J and we wanted to avoid introducing an additional difference between the state-space and the reference models.*

- Simulated discharge Q in Eq.2 is defined as an instantaneous flow I assume? Observed discharge is however an integrated quantity (total over an hour or a day). Wouldn't it be better to define simulated Q also as an integrated quantity? You could in fact add Eq.2 to the ODE system in Eq. 1: dQ/dt = Qr + Qd. Note that you then would have to reset Q = 0 at the start of each forcing time interval.

  You are right, the discharge presented by Eq.2 is an instantaneous flux. The simulated flow is the integration of this equation over the time-step. In the code, the integration is calculated using the adaptive sub-step implicit approximation. It can be seen in the "GR4_STSP.f" script (in the internal fluxes calculation part) of provided model sources. To clarify this point in the manuscript, we will add at line 7 page 6 (before the Eq.2) that the output equation is to calculate the instantaneous output flow $q(t)$. After this equation (where we will replace Q by $q(t)$) we will add that the simulated output Q is the integration of $q(t)$ over the time-step.

  **Added/Modified:** Sect. 2.4, p 8, line 20, *The output equation to calculate the instantaneous output flow ($q(t)$ in Eq. 10) completes the model:*

  $$q(t) = Q_r + Q_d$$

  Sect. 2.4, p 9, line 1, *Output: Q is the output flow, it corresponds to the integration of $q(t)$ (Eq. 10) over the time step.*

- It would be good to explicitly point out in table 1 that the instantaneous flux equations are the same for the two models.

  We agree and will point this out.

  **Added/Modified:** Table 1 title, *The discrete formulations are the continuous equations integrated individually over the modelling time step using the operator splitting technique while continuous equations correspond to the terms of the water balance differential equation of each store.*

5. Section 2: The discrete form is contrasted with the state-space form of the model. Note that a state-space representation can be either discrete or continuous, so it may be better to explicitly call it continuous state-space formulation.

   You are right, we will try to be more precise by writing, at least in section 2, that the state-space representation is continuous. However, because of the first point of the review, we will also mention a discrete (or operator-splitted) form of the state-space formulation.

**Added/Modified:** *Modified at different locations in the manuscript*

Sect. 3.3, p 13, line 1, *Three versions of the model were assessed on the 240 catchments following a split-sample test (Klemes, 1986). These three versions are the reference model, a discrete state-space model (with a Nash Cascade but solved using operator splitting) and a continuous state-space model.*

6. Section 4.3: this section describes the relation between the unit hydrograph approach for routing in the old model and the Nash cascade representation in the new model; in my view this section really fits better in the methods section, for example following the text at the bottom of page 6. My suggestion is to move it there.

   To be more comprehensive, we will try to add this in the section 2. Because of the first comment we will also mention the operator-splitted state-space model with the Nash Cascade and the continuous state-space formulation of this model in section 2.

   **Added/Modified:** *Done, moved to Sect. 2.3*

7. Abstract: what do you mean by "resolution"?

   By "resolution" we meant "solution". It will be fixed.

   **Added/Modified:** Abstract, *As a result, only the solutions of the split equations are used to present the different models.*

8. These typo mistakes will be corrected.

   **Added/Modified:** *Done*

Léonard Santos, on behalf of co-authors

[revised manuscript text omitted]